# Modes of Vibration in Basketball Rims and Backboards and the Energy Rebound Testing Device

**Daniel Winarski [1,*]** , **Kip P. Nygren [2]** and **Tyson Winarski [3]**

1 Independent Researcher, Tucson, AZ 85710, USA
2 Independent Researcher, Wilmington, NC 28405, USA; kip.nygren@gmail.com
3 Sandra Day O'Connor College of Law, Phoenix Campus, Arizona State University, Phoenix, AZ 85004, USA; tyson.winarski@asu.edu
* Correspondence: winarskifirm@gmail.com

**Abstract:** Six mode shapes, including bending and torsion, were documented for five different basketball rims and backboards at the United States Military Academy, West Point, New York, NY, USA. The frequency and damping ratio of each mode shape were also determined. The empirical process began with the time-domain excitation and response of each rim-backboard system. The impulse of excitation came from an impact hammer separately applied sequentially to each node. The sinusoidal response was gathered from an accelerometer at a fixed location (node 1). Each time-domain excitation response was then converted to a frequency-domain Bode plot for each node by a Brüel & Kjær 2034 Signal Analyzer, giving transfer functions of output/input versus frequency. Structural Measurements System (SMS) StarStruc software was used to fit mode shapes to the Bode plots. Each of the six mode shapes was fitted to the Bode plots of each node at a specific modal frequency. Each of the six mode shapes was a function of the locations of the nodes, and the Bode plots gathered at each node. The first and second modes were critical for showing that the Energy Rebound Testing Device statistically correlated with the energy transferred to the rim and backboard. A known perturbation mass was selectively attached to the rim to help isolate the dynamic masses and spring rates for the rim and backboard and to ascertain that the kinetic energy transferred to the rim had a 95.67% inverse correlation with rim stiffness.

**Keywords:** basketball rim and backboard; modal analysis; frequency; damping

## 1. Introduction

Javorski [1] characterized the dynamic behavior of a ceiling-mounted basketball goal using an impact hammer and a fixed-location accelerometer. Vibration measurements were taken at fourteen nodes, ten of which were on the frame supporting the basketball rim and backboard. Only four nodes were measured on the backboard, at the corners of the backboard, and none on the rim itself. Overall, 36 frequency response functions were measured, and this study concentrated mostly on structural vibrations between 0 and 10 Hz. Thus, this study was focused on the structural support of the backboard and rim rather than the elastic vibrations of the backboard and rim themselves.

Our study focused on the vibration modes of the elastic basketball backboard and rim. Like the Javorski study, we used an impact hammer and a fixed-location accelerometer. The 38-node no-shot-clock model used 30 measurement nodes to model the backboard, and we treated the backboard as a plate with vibratory motion perpendicular to the plane of that plate. 8 nodes were used to model the basketball rim. Two rim nodes and two backboard nodes were used to model the bracket for mounting the rim to the backboard. The 54-node shot-clock model increased the number of rim nodes from 8 to 16, increased the backboard nodes from 30 to 32, and used 6 nodes to model the support frame for the shot-clock. Two rim nodes and four backboard nodes were used to model the bracket for mounting the rim

to the backboard. We collected frequency response functions between 0 and 200 Hz, which included six bending and torsional modes of vibration of the basketball rim and backboard between 0 and 100 Hz.

Our primary goal was a statistical cross-correlation between the kinetic energy transfer reading of the Energy Rebound Testing Device [2,3] and the spring rate of the basketball rim. Using a two-degree-of-freedom lumped-parameter spring-mass system, augmented with a known perturbation mass, allowed us to isolate the spring rate of the basketball rim of four different basketball rim-backboard systems. Two of these basketball rim-backboard systems were ceiling-mounted, like in the Javorski study. The proper orthogonal decomposition (POD), also known as the Karhunen–Loève decomposition, as described by Feeney and Kappagantu [4], was very helpful in fitting our two-degree-of-freedom lumped-parameter model to the first two mode shapes of the basketball rim and backboard and in understanding the eigenvectors of these two mode shapes. The statistical correlation between energy absorbed readings from the ERTD and the spring rate of the rim concluded that the ERTD statistically correlated at R = 95.67% with rim stiffness, and hence rim elasticity, over a 35.3 to 58.2% energy absorption range. Thus, we concluded that the ERTD was indeed a viable means of testing basketball rims and backboards to help add consistency to the physics of this sport.

Looking beyond our first two modes of vibration, Dumond [5] provided valuable insight for modes 5–6, and the indicial notation used in this article was adopted for the quantification of plate-vibration-dominated modes 3–6. Irvine provided valuable insight for visualizing mode 4 [6] and dome-shaped mode 5 [7], especially since the plates that Irvine analyzed had the same aspect ratio of 1.5:1 as the basketball backboard in this study. Anđelić [8] and Guguloth [9] also provided visualization of modes 5–6.

Other studies involving the sport of basketball included Okubo and Hubbard [10,11], who analyzed the dynamics of basketball-rim interactions by using nonlinear ordinary differential equations to describe three components of ball angular velocity and contact point position on the toroidal rim. The rim and backboard were assumed to be rigid in this study. Russel [12] modeled basketballs as spherical acoustic cavities. Gharaibeh [13] was also helpful in understanding the higher-mode plate vibrations of the backboard. Oey [14] published MATLAB code for visualizing plate vibrations.

## 2. Materials and Methods

The process diagram for the Structural Measurements System (SMS) StarStruc software is shown in Figure 1. This StarStruc software ran on a portable computer.

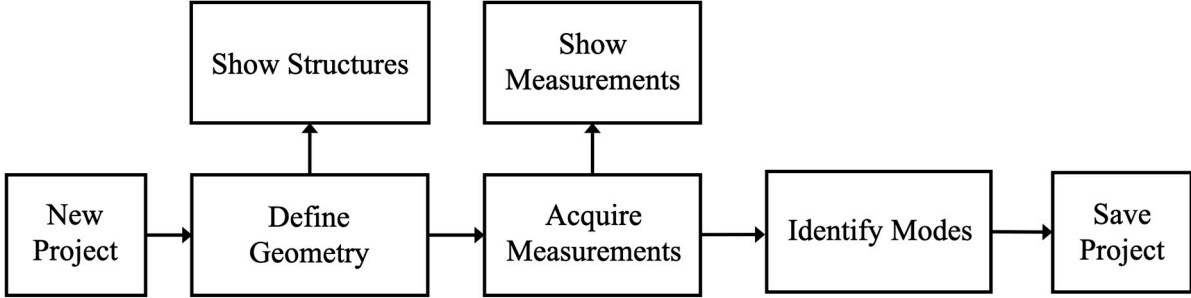

**Figure 1.** Process diagram for SMS StarStruc software.

Each of the rim-backboard systems without a shot-clock had the 38 nodes shown in Figure 2. The Cartesian coordinates of the 38 nodes were declared in Define Geometry and are listed in Table 1. The origin of the Cartesian coordinate system was the center of the circular rim, 381 mm (15 inches) from the backboard. The first 8 nodes were laid out counterclockwise, in an octagonal pattern, to document the 457 mm (18 inch) internal-diameter circular rim [15,16]. The circular rim comprised a 15.9 mm (5/8 inch) diameter circular torus with a mass of 2.3 kg. The remaining 30 nodes documented the backboard.

A steel bracket, defined by nodes 4-9-16-5, was used to attach the steel rim to the glass backboard. The layout of the grid on each actual rim and backboard was very tedious and usually took more time than the gathering of the excitation-response measurements. A large T-square used in mechanical drawing proved very helpful.

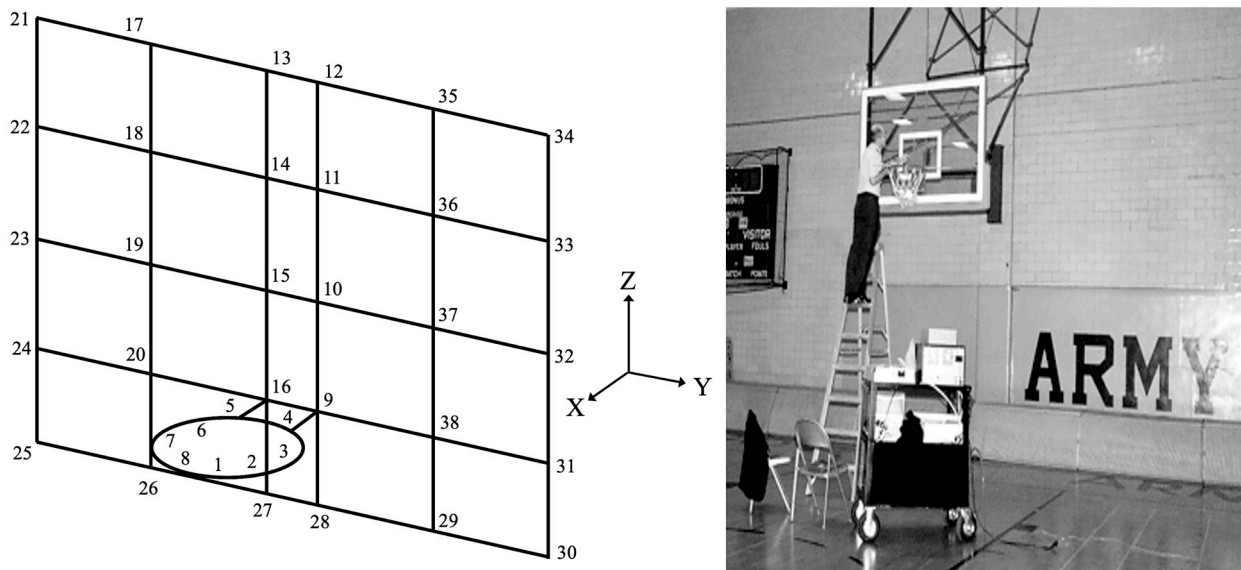

**Figure 2.** Layout of rim-backboard nodes, numbered 1–38 and defined in Table 1.

**Table 1.** Thirty-eight nodes of the basketball rim and backboard; coordinates in millimeters.

| Node | X | Y | Z | Component | Node | X | Y | Z | Component |
|---|---|---|---|---|---|---|---|---|---|
| 1 | 218 | 90 | 0 | Rim + 4393 | 2 | 90 | 218 | 0 | Rim |
| 3 | −90 | 218 | 0 | Rim | 4 | −218 | 90 | 0 | Rim |
| 5 | −218 | −90 | 0 | Rim | 6 | −90 | −218 | 0 | Rim |
| 7 | 90 | −218 | 0 | Rim | 8 | 218 | −90 | 0 | Rim |
| 9 | −381 | 90 | 0 | Backboard | 10 | −381 | 90 | 290 | Backboard |
| 11 | −381 | 90 | 580 | Backboard | 12 | −381 | 90 | 870 | Backboard |
| 13 | −381 | −90 | 870 | Backboard | 14 | −381 | −90 | 580 | Backboard |
| 15 | −381 | −90 | 290 | Backboard | 16 | −381 | −90 | 0 | Backboard |
| 17 | −381 | −478 | 870 | Backboard | 18 | −381 | −478 | 580 | Backboard |
| 19 | −381 | −478 | 290 | Backboard | 20 | −381 | −478 | 0 | Backboard |
| 21 | −381 | −868 | 870 | Backboard | 22 | −381 | −868 | 580 | Backboard |
| 23 | −381 | −868 | 290 | Backboard | 24 | −381 | −868 | 0 | Backboard |
| 25 | −381 | −868 | −250 | Backboard | 26 | −381 | −478 | −250 | Backboard |
| 27 | −381 | −90 | −250 | Backboard | 28 | −381 | 90 | −250 | Backboard |
| 29 | −381 | 478 | −250 | Backboard | 30 | −381 | 868 | −250 | Backboard |
| 31 | −381 | ,868 | 0 | Backboard | 32 | −381 | 868 | 290 | Backboard |
| 33 | −381 | 868 | 580 | Backboard | 34 | −381 | 868 | 870 | Backboard |
| 35 | −381 | 478 | 870 | Backboard | 36 | −381 | 478 | 580 | Backboard |
| 37 | −381 | 478 | 290 | Backboard | 38 | −381 | 478 | 0 | Backboard |

The 38 nodes in Table 1 had to be properly sequenced to display the geometry shown in Figure 1. This sequencing, also performed in Define Geometry, is shown in Table 2. The action of lifting the pen, designated by "×," was performed to avoid unwanted diagonal lines. Once Table 2 was completed, the rim-backboard shown in Figure 2 was displayed via Show Structures in Figure 1. As the rim-backboard was being assembled, Show Structures was periodically accessed to detect any mistakes before they pervasively propagated.

Once Define Structure was completed, the process went to Acquire Measurements (Figure 1). The instrumentation used in Acquire Measurements is shown in Figure 3. The

Brüel & Kjær 2644 line-driver charge amplifiers [17] were used to convert low-level signals from the Brüel & Kjær 8200 force transducer [18] and the Brüel & Kjær 4393 accelerometer [19]. The Brüel & Kjær 2644 charge amplifiers were always used together for both channel A (excitation) and channel B (response) when using the impact hammer. The orientation of the Brüel & Kjær 2644 was critical. The externally threaded end of each Brüel & Kjær 2644 had to be pointed towards the Brüel & Kjær 2034 analyzer, or no measurements could be taken.

The Brüel & Kjær 8202 impact hammer, Nærum, North Denmark, Denmark, (excitation) was used to individually gently tap each of the 38 nodes of Table 1 in succession, as outlined by Kuttner [20] and Irvine [21]. Rim nodes 1–8 were struck in the -Z direction, and backboard nodes 9–38 were struck in the -X direction. The fixed location of the Brüel & Kjær 4393 accelerometer (response) was node 1, and the accelerometer was oriented in the vertical +Z direction as declared in New Project, Figure 1. The Brüel & Kjær 4393 accelerometer was adhered to the rim via beeswax.

**Table 2.** Display sequence of 54 lines connecting 38 nodes of basketball rim and backboard.

| Line | Lift Pen | Start Node | End Node | Line | Lift Pen | Start Node | End Node | Line | Lift Pen | Start Node | End Node |
|---|---|---|---|---|---|---|---|---|---|---|---|
| 1 | × | 1 | 8 | 2 |  | 1 |  | 3 | × | 4 |  |
| 4 |  | 9 | 16 | 5 |  | 16 |  | 6 |  | 5 |  |
| 7 | × | 9 |  | 8 |  | 16 |  | 9 | × | 10 |  |
| 10 |  | 15 |  | 11 | × | 11 |  | 12 |  | 14 |  |
| 13 | × | 17 | 20 | 14 | × | 21 | 38 | 15 | × | 13 |  |
| 16 |  | 17 |  | 17 | × | 14 |  | 18 |  | 18 |  |
| 19 | × | 15 |  | 20 |  | 19 |  | 21 | × | 16 |  |
| 22 |  | 20 |  | 23 | × | 17 |  | 24 |  | 21 |  |
| 25 | × | 18 |  | 26 |  | 22 |  | 27 | × | 19 |  |
| 28 |  | 23 |  | 29 | × | 20 |  | 30 |  | 24 |  |
| 31 | × | 20 |  | 32 |  | 26 |  | 33 | × | 16 |  |
| 34 |  | 27 |  | 35 | × | 9 |  | 36 |  | 28 |  |
| 37 | × | 38 |  | 38 |  | 29 |  | 39 | × | 9 |  |
| 40 |  | 38 |  | 41 | × | 10 |  | 42 |  | 37 |  |
| 43 | × | 11 |  | 44 |  | 36 |  | 45 | × | 12 |  |
| 46 |  | 35 |  | 47 | × | 34 |  | 48 |  | 35 |  |
| 49 | × | 33 |  | 50 |  | 36 |  | 51 | × | 32 |  |
| 52 |  | 37 |  | 53 | × | 31 |  | 54 |  | 38 |  |

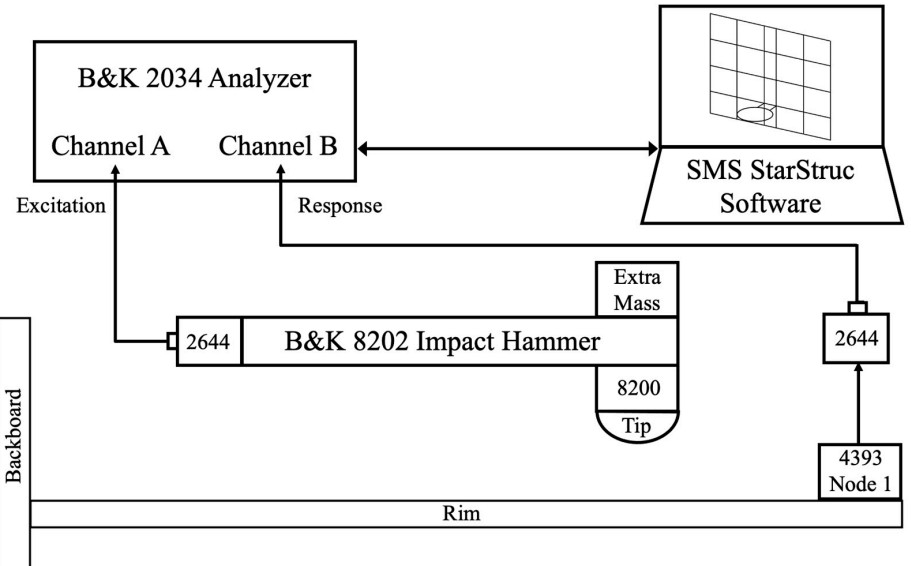

**Figure 3.** The Brüel & Kjær 8202 impact hammer and 4393 accelerometer instrumentation.

Table 3 shows six possible configurations of the Brüel & Kjær 8202 impact hammer, which could be used to adjust the frequency range of the excitation. Three tips were available: steel, plastic, and hard rubber. The softer the tip, the more the upper frequency of excitation was attenuated. A further reduction in the upper frequency of excitation could be achieved by adding an extra mass. For this study, the hard rubber tip and the extra mass were used to limit the excitation to 0–340 Hz. We had no interest in vibrations in the kHz range, so we did not excite the rim-backboard system at that level. Additionally, we did not want any damage to the rim or backboard, perceived or actual, so the use of the steel tip was never considered.

**Table 3.** Frequency ranges of excitation for various im-pact hammer configurations.

| Tip Attached to Impact Hammer | No Extra Mass | Extra Mass |
|---|---|---|
| Steel | 0–7 kHz | 0–4.5 kHz |
| Plastic | 0–2 kHz | 0–1.3 kHz |
| Hard Rubber | 0–500 Hz | 0–340 Hz |

The use of the Brüel & Kjær 8202 impact hammer was part art and part science. The goal was to impart a single-hit impulse at each node in Table 1. Then the response to the impulse at that node was measured by the Brüel & Kjær 4393 accelerometer at node 1. It was critical that the impulse hammer never have a double-hit, as such a double-hit would have made it impossible to gather the desired output/input Bode plot. Thus, there was a certain amount of art in the wrist action of the user to impart a single-hit impulse that was not too severe yet not too soft. One subjective clue as to the adequacy of the application of the impact hammer was the low-frequency sound made by the rim-backboard. Thus, aural feedback was very important.

Table 4 gives the windowing and gains assigned to the Brüel & Kjær 8202 impact hammer and the Brüel & Kjær 4393 accelerometer.

**Table 4.** Windowing and gains assigned to the Brüel & Kjær 8202 impact hammer and 4393 accelerometer.

| Instrumentation | 8202 Impact Hammer | 4393 Accelerometer |
|---|---|---|
| B&K 2034 Channel | A | B |
| Window | Rectangular | Rectangular |
| Gain | 1.01 mV/N | 318 $\mu$V/m/s$^2$ |

Once each of the 38 nodes was struck five times by the Brüel & Kjær 8202 impact hammer, to average out noise, the excitation-response data gathered by the Brüel & Kjær 2034 Analyzer was stored as a *.FRF (frequency response function) file in the portable computer by the SMS StarStruc software. It was important to have a separate project for each rim-backboard so that new *.FRF files for one project did not overlay previously measured *.FRF files for another project.

Figure 4 shows the magnitude versus frequency of a typical Bode plot of excita-tion/response. Near the bottom of Figure 4, rectangular windows were shown to be de-clared, per Table 4. At the bottom right corner of Figure 4, the settings of 1.01 mV/N for channel A (Brüel & Kjær 8202 impact hammer) and 318 $\mu$V/m/s$^2$ for channel B (Brüel & Kjær 4393 accelerometer) are shown.

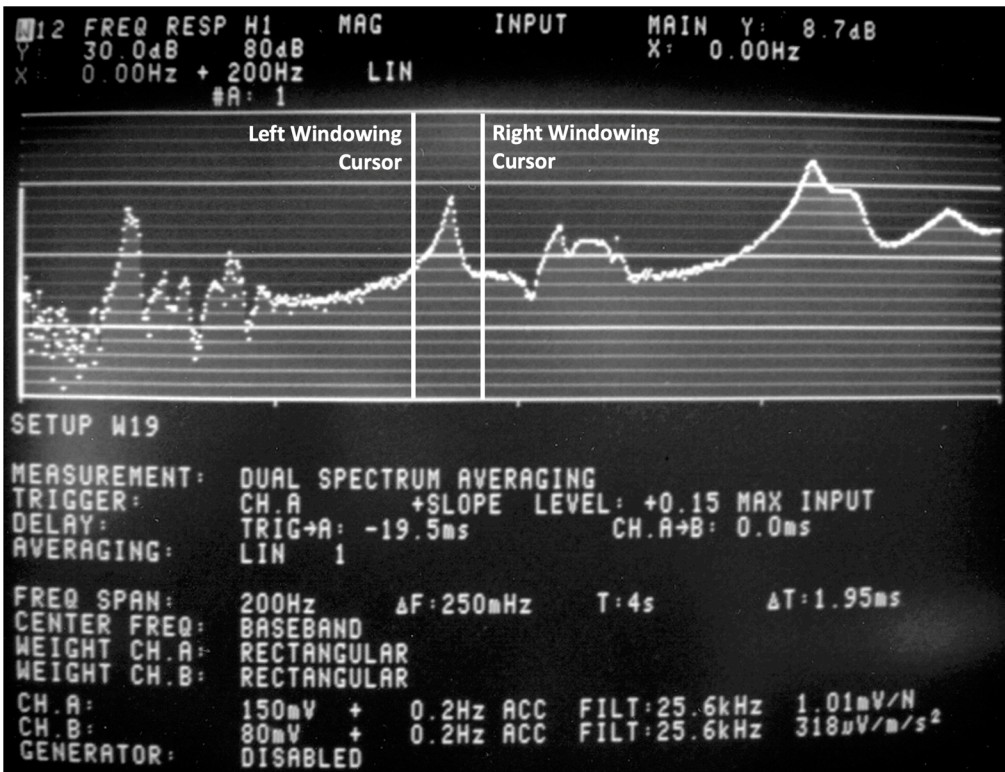

**Figure 4.** Magnitude versus frequency of a typical bode plot of Excitation/Response.

Once all of the bode plots were calculated for the 38 nodes in Table 1, SMS StarStruc software was used to fit mode shapes to the bode plots using the polynomial method. To fit each mode shape, a pair of windowing cursors were used to manually bracket clearly discernible peaks in a Bode plot for the processing of a mode shape attributed to that peak in magnitude, an example of which is shown in Figure 4.

Table 5 shows that there were six peaks of interest below 100 Hz for the rim-backboard without a shot-clock. We took advantage of the band option of the SMS StarStruc software, which allowed the identification of multiple modes within one band. The approximate left and right cursor locations used to bracket each peak are listed in Table 5.

The six modes identified by using Table 5 are shown in Figures 5–10 for the case of a Hydra-Rib rim and backboard without a shot-clock. Both vector and contour plots are shown for each mode shape to assist the reader. The first two modes of a Hydra-Rib with a shot-clock are shown in Figures 11 and 12. The modal frequency and damping are listed for each mode shape. Siemens gave an instructive tutorial on mode identification [22]. Bold arrows are used to denote eigenvectors of the motion of the rim relative to the backboard, as well as the plate vibrations of the backboard.

**Table 5.** Six peaks of interest in bode plots with corresponding left–right cursor locations.

| Approximate Peaks | Band | Windowing Cursors | Low Mode # | Number of Modes |
|---|---|---|---|---|
| 24, 33, 41 Hz | 1 | 15.5–46.75 Hz | 1 | 3 (Figures 6–8) |
| 51 Hz | 2 | 48–55 Hz | 4 | 1 (Figure 9) |
| 78 and 94 Hz | 3 | 72.25–99.75 Hz | 5 | 2 (Figures 9 and 10) |

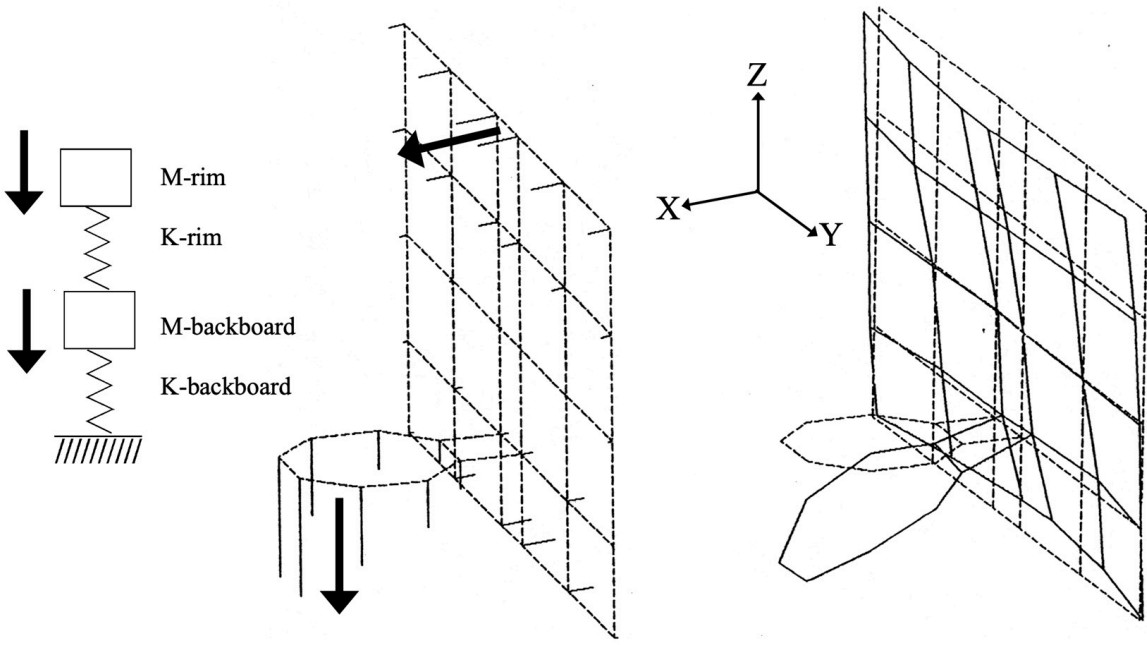

**Figure 5.** Mode-1: $\omega$ = 23.62 Hz; damping ratio $\zeta$ = 2.66%.

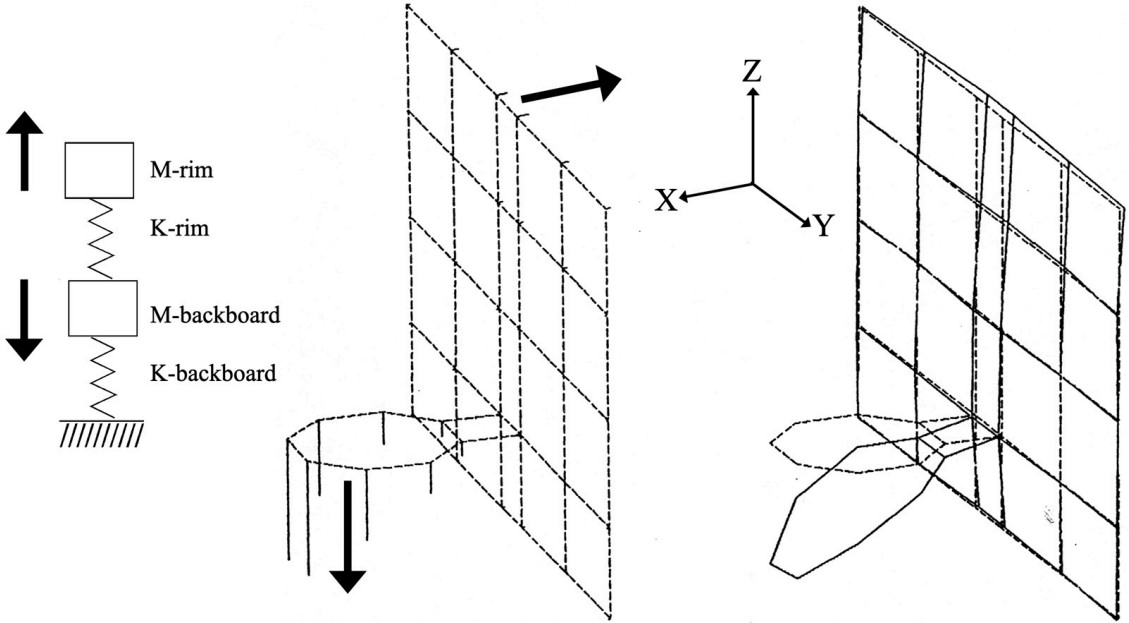

**Figure 6.** Mode-2: $\omega$ = 33.08 Hz; damping ratio $\zeta$ = 2.55%.

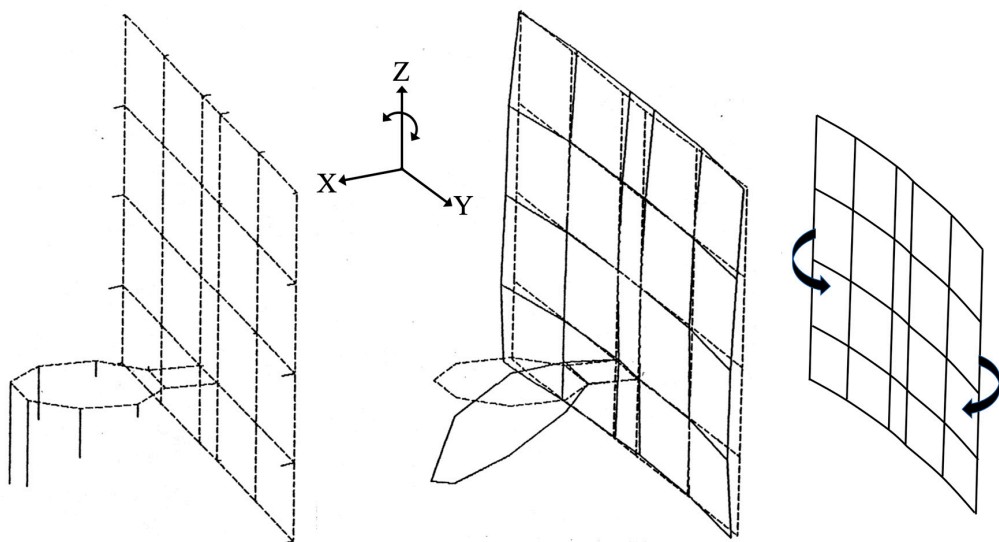

**Figure 7.** Mode-3 (my = 0, mz = 1): ω = 41.54 Hz; damping ratio ζ = 3.69%.

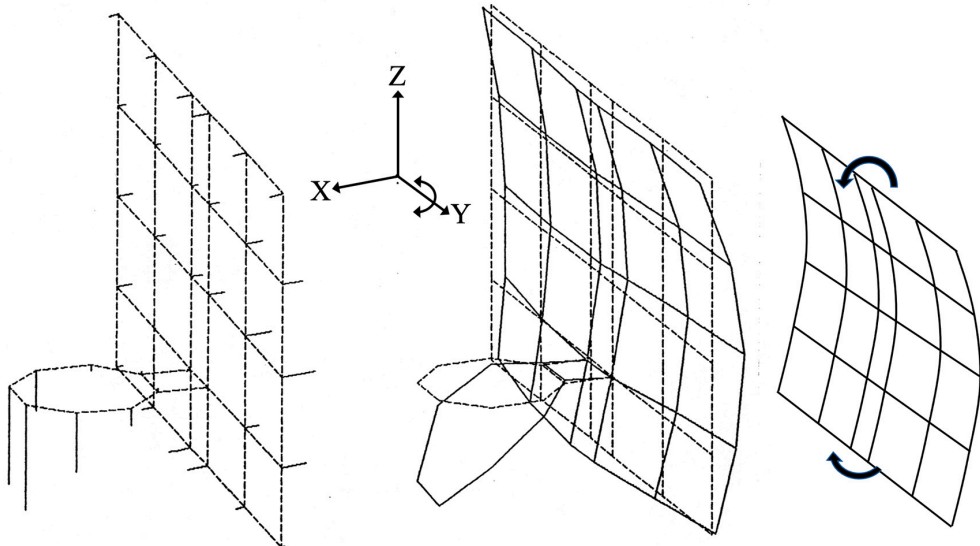

**Figure 8.** Mode-4 (my = 1, mz = 0): ω = 51.45 Hz; damping ratio ζ = 1.31%.

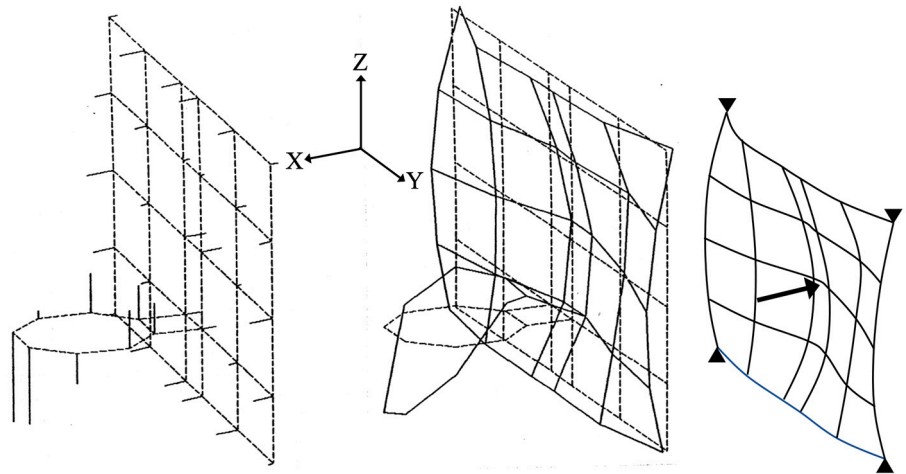

**Figure 9.** Mode-5 (my = 1, mz = 1): ω = 78.14 Hz; damping ratio ζ = 1.46%.

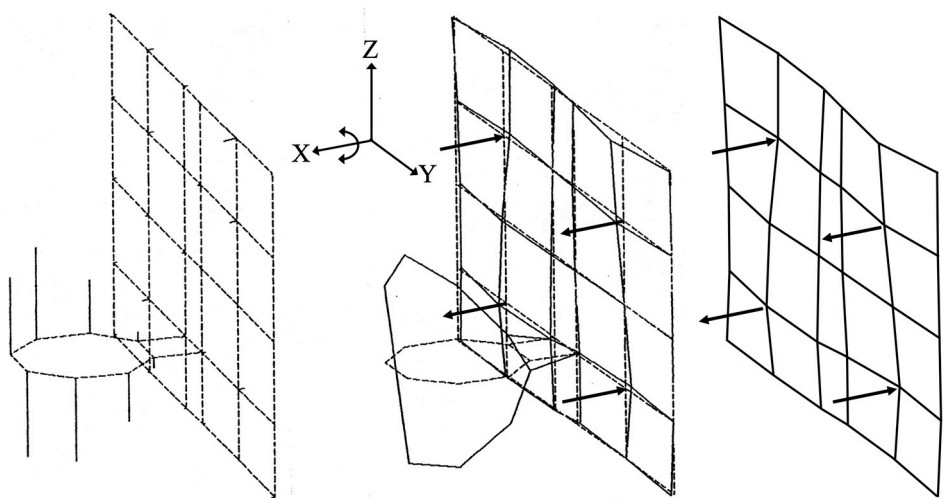

**Figure 10.** Mode-6 (my = 2, mz = 2): ω = 94.38 Hz; damping ratio ζ = 2.85%.

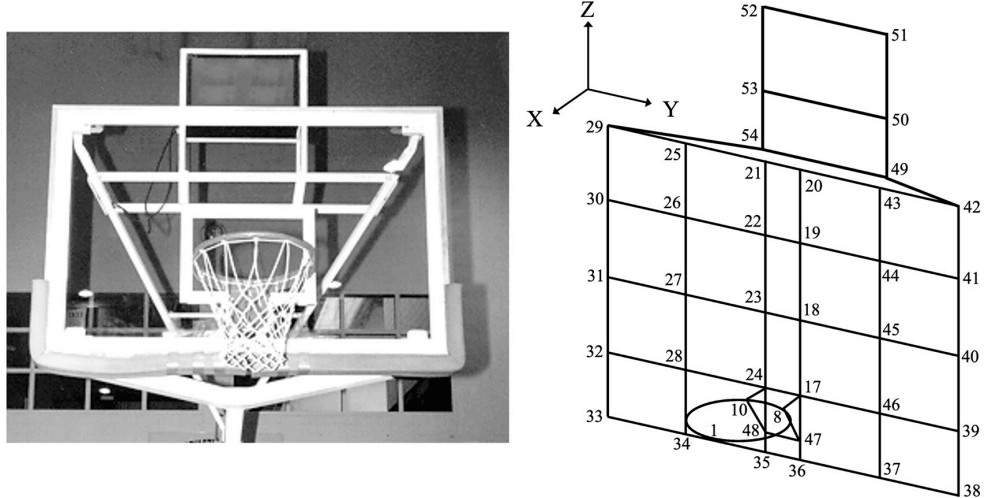

**Figure 11.** Hydra-Rib basketball rim, backboard, and shot-clock, nodes numbered 1–54.

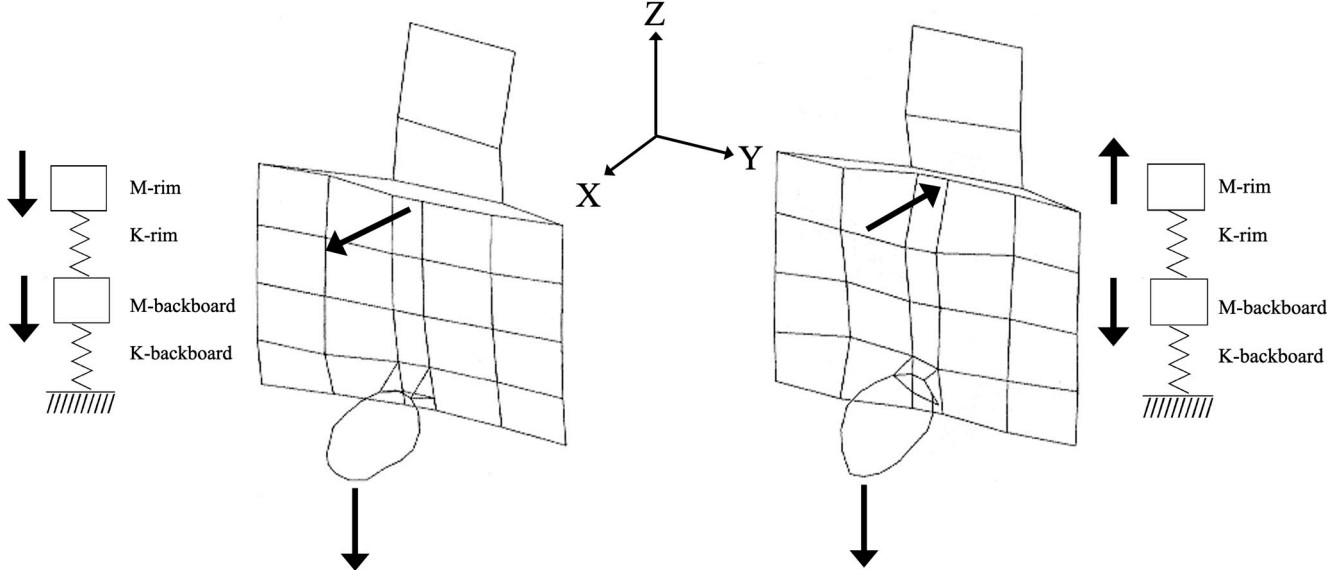

**Figure 12.** Hydra-Rib with shot-clock: Mode 1 (24.72 Hz) and Mode 2 (29.93 Hz).

### 3. Results

The first two rim-backboard modes, Figures 5 and 6, were equivalent to a two-mass, two-spring system. In Figure 5, the rim and backboard were in phase, as shown by the bold arrow eigenvectors pointing in the same direction, thus representing the lowest modal frequency of 23.62 Hz. In Figure 6, the rim and backboard were 180° out of phase, as shown by the bold arrow eigenvectors pointing opposite directions, thus increasing the modal frequency to 33.08 Hz. These two modes correspond exactly to the eigenvectors of a 2-spring, 2-mass system, as shown on the left of Figures 5 and 6.

A lot was happening in Figures 7 and 8. However, a first-order approximation is that modes 3 and 4 were influenced by the backboard plate bending as a simple plane. In Figure 7, the plane of the backboard was bending about the Z (vertical) axis ($my = 0$, $mz = 1$). However, in Figure 8, the plane of the backboard was bending as the Y (horizontal) axis ($my = 1$, $mz = 0$). Since the Y dimension of the backboard was 1.83 m and the Z dimension was shorter at 1.22 m, per Table 6, the modal frequency of 41.54 Hz in Figure 7 is lower than the modal frequency of 51.45 Hz in Figure 8. The circular rim (nodes 1–8) appeared to be hinging where it attached to the steel mount bracket (nodes 4–9–16–5).

Table 6 gives the parameters of the tempered glass used in each Hydra-Rib backboard. The length and width measurements of the rectangular tempered glass included a surrounding frame.

**Table 6.** Tempered glass in Hydra-Rib backboards.

| Youngs Modulus: 69 GPa | Areal Density: 31 kg/m$^2$ | Thickness (X): 12.7 mm |
|---|---|---|
| Length (Y): 1.83 m | Height (Z): 1.22 m | Aspect Ratio (Y:Z): 1.5:1 |

In Figure 9, the backboard flexed in a dome-like deformation along the X direction ($my = 1$, $mz = 1$) at a higher frequency, 78.14 Hz. This dome-like deformation of the backboard is similar to Irvine's Figure 3 [7] for a plate point supported at each corner, which also had an aspect ratio of 1.5:1. The same plate point-supported boundary condition was consistent with modes 3–4, in Figures 7 and 8, above. The backboard appeared to have its four corners constrained by the Hydra-Rib mount. The rim itself was now flexing in a more complicated mode shape, similar to the second mode of flexural vibration of a beam.

Figure 10 exhibits the first torsion of the rim about the *X*-axis, coupled with a higher mode of vibration for the backboard plate ($my = 2$, $mz = 2$). This was at the highest modal frequency, 94.38 Hz, which we pursued.

The modal frequencies and damping ratios in Figures 5–10 are summarized in the left frequency-damping data column in Table 7 below. All six modes of the rim-backboard were lightly damped, with the damping ratio ranging between $0.46\% \leq \zeta \leq 5.21\%$. Thus, the damped natural frequencies and the natural frequencies were essentially equal.

**Table 7.** Summary of the six modes, their frequencies $\omega$, and damping ratios $\zeta$.

| Backboard | Hydra-Rib (Figures 5–10) | Hydra-Rib | Ceiling | Ceiling | Hydra-Rib + Shot |
|---|---|---|---|---|---|
| Mode 1 | $\omega = 23.62$ Hz $\zeta = 2.66\%$ | $\omega = 26.48$ Hz $\zeta = 1.77\%$ | $\omega = 22.24$ Hz $\zeta = 0.90\%$ | $\omega = 22.77$ Hz $\zeta = 1.98\%$ | $\omega = 24.72$ Hz $\zeta = 0.90\%$ |
| Mode 2 | $\omega = 33.08$ Hz $\zeta = 2.55\%$ | $\omega = 36.68$ Hz $\zeta = 2.48\%$ | $\omega = 33.98$ Hz $\zeta = 5.21\%$ | $\omega = 34.48$ Hz $\zeta = 3.04\%$ | $\omega = 29.93$ Hz $\zeta = 2.60\%$ |
| Mode 3 | $\omega = 41.54$ Hz $\zeta = 3.69\%$ | $\omega = 53.01$ Hz $\zeta = 0.78\%$ | $\omega = 53.23$ Hz $\zeta = 1.72\%$ | $\omega = 56.30$ Hz $\zeta = 0.68\%$ | $\omega = 39.47$ Hz $\zeta = 1.14\%$ |
| Mode 4 | $\omega = 51.45$ Hz $\zeta = 1.31\%$ | $\omega = 58.31$ Hz $\zeta = 0.88\%$ | $\omega = 57.93$ Hz $\zeta = 0.83\%$ | $\omega = 61.74$ Hz $\zeta = 1.79\%$ | $\omega = 42.93$ Hz $\zeta = 0.78\%$ |
| Mode 5 | $\omega = 78.14$ Hz $\zeta = 1.46\%$ | $\omega = 76.03$ Hz $\zeta = 4.59\%$ | $\omega = 67.56$ Hz $\zeta = 0.46\%$ | $\omega = 74.10$ Hz $\zeta = 0.65\%$ | $\omega = 51.32$ Hz $\zeta = 2.05\%$ |
| Mode 6 | $\omega = 94.38$ Hz $\zeta = 2.85\%$ | $\omega = 81.45$ Hz $\zeta = 0.57\%$ | $\omega = 81.03$ Hz $\zeta = 0.63\%$ | $\omega = 85.82$ Hz $\zeta = 0.68\%$ | $\omega = 59.48$ Hz $\zeta = 1.23\%$ |

This study then included the Hydra-Rib basketball rim and backboard with a shot-clock, as shown in Figure 11.

In Table 8, six additional nodes (49–54) were needed to add the shot-clock, and the steel bracket connecting the steel rim and glass backboard was augmented with two additional nodes (47–48). The rim (now nodes 1–16) was augmented with eight additional nodes to model the rim as a sixteen-sided hexadecagon (previously an octagon).

**Table 8.** Fifty-four nodes of the basketball rim, backboard, and shot-clock (in millimeters).

| Node | X | Y | Z | Component | Node | X | Y | Z | Component |
|---|---|---|---|---|---|---|---|---|---|
| 1 | 236 | 0 | 0 | Rim + 4393 | 2 | 218 | 90 | 0 | Rim |
| 3 | 167 | 167 | 0 | Rim | 4 | 90 | 218 | 0 | Rim |
| 5 | 0 | 236 | 0 | Rim | 6 | −90 | 218 | 0 | Rim |
| 7 | −167 | 167 | 0 | Rim | 8 | −218 | 90 | 0 | Rim |
| 9 | 236 | 0 | 0 | Rim | 10 | −218 | −90 | 0 | Rim |
| 11 | 167 | −167 | 0 | Rim | 12 | −90 | −218 | 0 | Rim |
| 13 | 0 | −236 | 0 | Rim | 14 | 90 | −218, | 0 | Rim |
| 15 | 167 | −167 | 0 | Rim | 16 | 218 | −90, | 0 | Rim |
| 17 | −381 | 90 | 0 | Backboard | 18 | −381 | 90 | 290 | Backboard |
| 19 | −381 | 90 | 580 | Backboard | 20 | −381 | 90 | 870 | Backboard |
| 21 | −381 | −90 | 870 | Backboard | 22 | −381 | −90 | 580 | Backboard |
| 23 | −381 | −90 | 290 | Backboard | 24 | −381 | −90 | 0 | Backboard |
| 25 | −381 | −478 | 870 | Backboard | 26 | −381 | −478 | 580 | Backboard |
| 27 | −381 | −478 | 290 | Backboard | 28 | −381 | −478 | 0 | Backboard |
| 29 | −381 | −868 | 870 | Backboard | 30 | −381 | −868 | 580 | Backboard |
| 31 | −381 | −868 | 290 | Backboard | 32 | −381 | −868 | 0 | Backboard |
| 33 | −381 | −868 | −250 | Backboard | 34 | −381 | −478 | −250 | Backboard |
| 35 | −381 | −90 | −250 | Backboard | 36 | −381 | 90 | −250 | Backboard |
| 37 | −381 | 478 | −250 | Backboard | 38 | −381 | 868 | −250 | Backboard |
| 39 | −381 | 868 | 0 | Backboard | 40 | −381 | 868 | 290 | Backboard |
| 41 | −381 | 868 | 580 | Backboard | 42 | −381 | 868 | 870 | Backboard |
| 43 | −381 | 478 | 870 | Backboard | 44 | −381 | 478 | 580 | Backboard |
| 45 | −381 | 478 | 290 | Backboard | 46 | −381 | 478 | 0 | Backboard |
| 47 | −381 | 90 | −180 | Backboard | 48 | −381 | −90 | −180 | Backboard |
| 49 | −730 | 315 | 780 | Shot-Clock | 50 | −730 | 315 | 1070 | Shot-Clock |
| 51 | −730 | 315 | 1470 | Shot-Clock | 52 | −730 | −315 | 1470 | Shot-Clock |
| 53 | −730 | −315 | 1070 | Shot-Clock | 54 | −730 | −315 | 780 | Shot-Clock |

In Figure 11, there was simply not enough room to show all sixteen nodes comprising the circular rim. However, node 1, where the 4393 accelerometer was attached with beeswax, is shown at the very front end of the circular rim. Rim nodes 8 and 10 are also shown, because these six nodes (8–17–47–48–24–10) now comprise the steel bracket holding the steel rim (nodes 1–16) to the backboard.

The 54 nodes in Table 8 had to be properly sequenced to display the geometry shown in Figure 11. This sequencing, also performed in Define Geometry, is shown in Table 9. The three columns of Line Numbers in Table 9 are in bold to make the line definitions in Table 9 easier to read. The action of lifting the pen, designated by "×," was performed to avoid unwanted diagonal lines. Once Table 9 was completed, the rim-backboard shown in Figure 11 was displayed via Show Structures, Figure 1. As the rim-backboard was being assembled, Show Structures was periodically accessed to detect any mistakes before they pervasively propagated.

**Table 9.** Display sequence of 63 lines connecting 54 nodes of the rim, backboard, and shot-clock.

| Line | Lift Pen | Start Node | End Node | Line | Lift Pen | Start Node | End Node | Line | Lift Pen | Start Node | End Node |
|------|----------|-----------|----------|------|----------|-----------|----------|------|----------|-----------|----------|
| 1 | × | 1 | 16 | 2 | | 1 | | 3 | × | 8 | |
| 4 | | 17 | 24 | 5 | | 10 | | 6 | × | 17 | |
| 7 | | 24 | | 8 | × | 18 | | 9 | | 23 | |
| 10 | × | 19 | | 11 | | 22 | | 12 | × | 25 | 28 |
| 13 | × | 29 | 46 | 14 | × | 21 | | 15 | | 25 | |
| 16 | × | 22 | | 17 | | 26 | | 18 | × | 23 | |
| 19 | | 27 | | 20 | × | 24 | | 21 | | 28 | |
| 22 | × | 25 | | 23 | | 29 | | 24 | × | 26 | |
| 25 | | 28 | | 26 | × | 27 | | 27 | | 31 | |
| 28 | × | 28 | | 29 | | 32 | | 30 | × | 28 | |
| 31 | | 34 | | 32 | × | 24 | | 33 | | 35 | |
| 34 | × | 17 | | 35 | | 36 | | 36 | × | 46 | |
| 37 | | 37 | | 38 | × | 17 | | 39 | | 46 | |
| 40 | × | 18 | | 41 | | 45 | | 42 | × | 19 | |
| 43 | | 44 | | 44 | × | 20 | | 45 | | 43 | |
| 46 | × | 42 | | 47 | | 43 | | 48 | × | 41 | |
| 49 | | 44 | | 50 | × | 40 | | 51 | | 45 | |
| 52 | × | 39 | | 53 | | 46 | | 54 | × | 8 | |
| 55 | | 47 | | 56 | | 10 | | 57 | × | 42 | |
| 58 | | 49 | | 59 | | 29 | | 60 | × | 49 | |
| 61 | | 54 | | 62 | × | 50 | | 63 | | 53 | |

Figure 12 above shows the first and second modes of the Hydra-Rib with a shot-clock. Analogous to the first two modes shown in Figures 5 and 6, the two modes shown in Figure 12 were equivalent to a two-mass, two-spring system. In Figure 12, the rim and backboard were in phase for the first mode, as shown by the bold arrow eigenvectors pointing in the same direction, thus representing the lowest modal frequency of 24.72 Hz. For the second mode in Figure 12, the rim and backboard were 180° out of phase, as shown by the bold arrow eigenvectors pointing in opposite directions, thus increasing the modal frequency to 29.93 Hz. These data are summarized in the right-most column of Table 7 above.

Once the first six modes of vibration were understood for the rim-backboard, the decision was made to focus on the first two modes in order to isolate the rim stiffness by means of a perturbation mass $Mp$ hung from node 1 and compare that to the energy reading of the Energy Rebound Testing Device, ERTD. The ERTD used a dropped mass to measure the energy transferred to the rim.

The Fair-Court® Energy Rebound Testing Device (ERTD) mimicked dropping a basketball on the outer end of the rim. Figure 13 shows the Fair-Court® Energy Rebound Testing Device, which has a long rod with a hook on the upper end that removably fastens to the end of the basketball rim. Along the long rod is a stop, which the drop-mass is held against before the drop-mass makes a 0.76 m (30 inch) drop to the base of the ERDT. Within the base of the ERTD is a compression spring, which causes the drop-mass to rebound. Also within the base is a photo-sensor that detects the transit of a 100 mm-long highly reflective portion of the drop-mass both during its initial descent towards the compression spring and its subsequent rebound. The 100 mm distance $\Delta Z$ divided by the downward transit time $\Delta T1$ gave the drop velocity $\Delta Z/\Delta T1$. The same 100 mm distance $\Delta Z$ divided by the rebound transit time $\Delta T2$ gave the rebound velocity $\Delta Z/\Delta T2$. The ratio of the change in kinetic energy divided by the original kinetic energy was given by this expression: $[m \times \Delta Z/\Delta T1)^2 - m \times (\Delta Z/\Delta T2)^2]/[m \times (\Delta Z/\Delta T1)^2]$, where drop-mass m was 0.74 kg. Since drop-mass m and $\Delta Z^2$ occur in both the numerator and denominator, the kinetic energy ratio was simplified to $[1/\Delta T1^2 - 1/\Delta T2^2]/[1/\Delta T1^2]$, which was further simplified to $1 - (\Delta T1/\Delta T2)^2$, agreeing precisely with column 12 of the Abbott-Davis patent [3]. This

is the reading displayed by the ERTD, and it is a measure of the energy absorbed by the basketball rim and backboard.

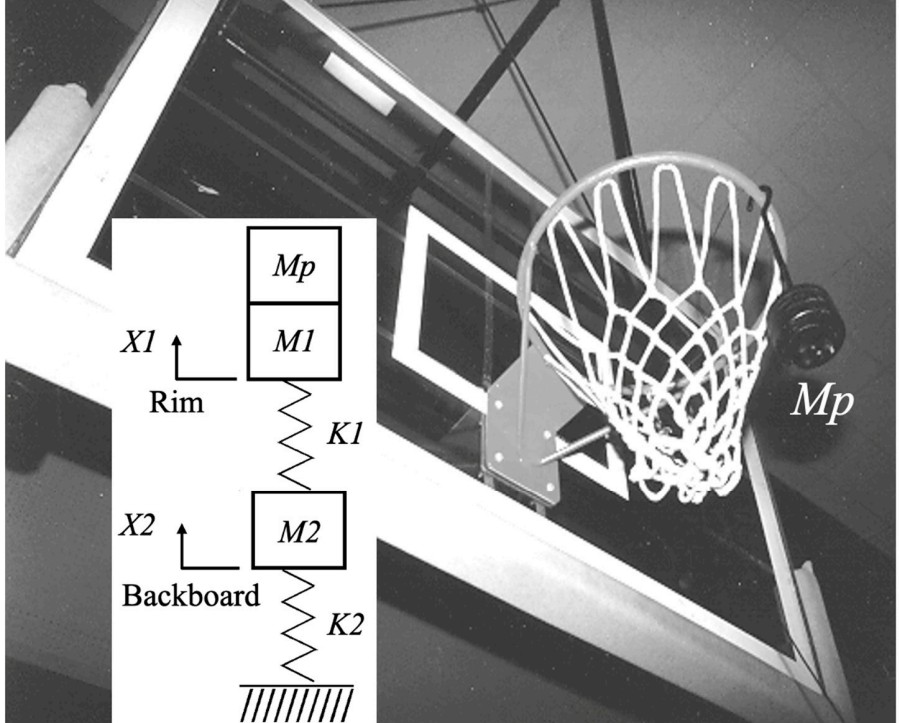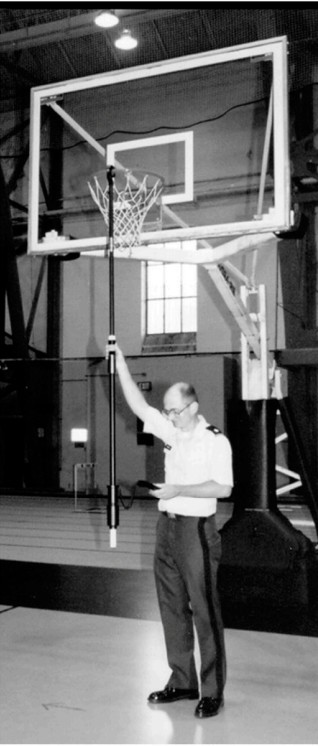

**Figure 13.** Perturbation mass (*Mp*) and energy rebound testing device (ERTD).

As shown in Figure 13, the first two modes of vibration were modeled as a two-spring, two-mass lumped parameter system. Masses *M1* and *M2* represent the dynamic masses, and spring rates *K1* and *K2* represent the dynamic spring rates of the rim and backboard, respectively, at node 1 (Figure 11). Determining *K1*, *M1*, *K2*, and *M2* required four equations for these four unknowns. The quadratic characteristic equations used to determine the eigenvalues $\lambda$ came from the following two degree-of-freedom differential equations of motion (2DOF DEOM), as described by Feeney and Kappagantu [4].

$$\begin{bmatrix} M1 + Mp & 0 \\ 0 & M2 \end{bmatrix} \begin{bmatrix} \ddot{X}1 \\ \ddot{X}2 \end{bmatrix} + \begin{bmatrix} K1 & -K1 \\ -K1 & K1 + K2 \end{bmatrix} \begin{bmatrix} X1 \\ X2 \end{bmatrix} = \begin{bmatrix} 0 \\ 0 \end{bmatrix}$$

The following determinant was used to find the quadratic expression for eigenvalues $\lambda$. This determinant comprised the inverse of the mass matrix times the spring matrix from the 2DOF DEOM, minus $\lambda$ times the identity matrix.

$$\det \begin{bmatrix} [K1/(M1 + Mp)] - \lambda & -K1/(M1 + Mp) \\ -K1/M2 & (K1 + K2)/M2 - \lambda \end{bmatrix} = 0$$

These eigenvalues were the square of the respective natural frequency in radians per second. The use of perturbation mass *Mp* provides two eigenvalue equations, and no perturbation mass (*Mp* = 0) provides the additional two eigenvalue equations needed.

$$\lambda^2 - \lambda \left[ K1/(M1 + Mp) + (K1 + K2)/M2 \right] + (K1 \times K2)/[(M1 + Mp) \times M2] = 0$$

Four natural frequencies, two modes with and two modes without a perturbation mass (*Mp*), were measured in Hertz via the modal analysis methods described above. These four natural frequencies were then converted to radians per second and squared to obtain the four eigenvalues used to calculate *K1*, *M1*, *K2*, and *M2*, Table 10.

**Table 10.** Calculation of spring rates and masses of rim and backboard and ERTD reading.

| Backboard | Hydra-Rib | Ceiling Mount | Ceiling Mount | Hydra-Rib + Shot Clock (Figure 12) |
|---|---|---|---|---|
| Mode 1 | $\omega$ = 26.48 H$\zeta$ | $\omega$ = 22.24 H$\zeta$ | $\omega$ = 22.77 H$\zeta$ | $\omega$ = 24.72 H$\zeta$ |
| Mode 2 | $\omega$ = 36.68 H$\zeta$ | $\omega$ = 33.98 H$\zeta$ | $\omega$ = 34.48 H$\zeta$ | $\omega$ = 29.93 H$\zeta$ |
| Perturbation Mass | $Mp$ = 0 kg | $Mp$ = 0 kg | $Mp$ = 0 kg | $Mp$ = 0 kg |
| Mode 1 | $\omega$ = 14.29 H$\zeta$ | $\omega$ = 15.25 H$\zeta$ | $\omega$ = 15.18 H$\zeta$ | $\omega$ = 11.30 H$\zeta$ |
| Mode 2 | $\omega$ = 30.06 H$\zeta$ | $\omega$ = 24.14 H$\zeta$ | $\omega$ = 25.35 H$\zeta$ | $\omega$ = 26.79 H$\zeta$ |
| Perturbation Mass | $Mp$ = 4.5 kg | $Mp$ = 3 kg | $Mp$ = 3 kg | $Mp$ = 4.5 kg |
| $K1$ – rim | 50,500 N/m | 40,000 N/m | 41,550 N/m | 28,300 N/m |
| $M1$ – rim | 1.1 kg | 0.925 kg | 0.96 kg | 0.9 kg |
| $K2$ – backboard | 568,000 N/m | 633,500 N/m | 480,500 N/m | 829,000 N/m |
| $M2$ – backboard | 17.8 kg | 30.7 kg | 21.7 kg | 30.5 kg |
| ERTD Reading | 35.30% | 40% | 42.10% | 58.20% |

The eigenvectors $\xi$ associated with the 2DOF DEOM were found to be very interesting and were derived using the following matrix equation. To compare these eigenvectors with Figures 5 and 6, the perturbation mass $Mp$ was set to zero.

$$\begin{bmatrix} K1/M1 - \lambda & -K1/M1 \\ -K1/M2 & (K1+K2)/M2 - \lambda \end{bmatrix} \{\xi\} = 0$$

Using the Hydra-Rib without a shot-clock in Table 10 as an example, the eigenvalue calculated for the first mode was $\lambda$ = 27,624 radians$^2$/second$^2$. With $K1$, $M1$, $K2$, and $M2$ defined in Table 10, the matrix equation for the first eigenvector $\{\xi1\}$ became the following:

$$\begin{bmatrix} 18285 & -45909 \\ -2837 & 7123 \end{bmatrix} \{\xi1\} = 0, \text{ the solution of which was } \{\xi1\} = \begin{Bmatrix} 0.929 \\ 0.370 \end{Bmatrix}.$$

Thus, the calculated eigenvector $\{\xi1\}$ for mode 1 shows the motion of the rim and backboard to be in phase, exactly as shown in empirical Figure 5. Furthermore, the ratio of the rim-to-backboard motion for mode-1 was calculated as 0.929/0.370 = 2.51. This calculated ratio of the rim-to-backboard motion of 2.51 was reasonably close to the empirical ratio of the rim-to-backboard motion of approximately 2.94, as measured by using mechanical calipers and zooming in on the maximum displacement vectors shown in Figure 5.

The eigenvalue calculated for the second mode was $\lambda$ = 53,032 radians$^2$/second$^2$. The matrix equation for the second eigenvalue $\{\xi2\}$ became the following:

$$\begin{bmatrix} 18285 - 7123 & -45909 \\ -2837 & -18285 \end{bmatrix} \{\xi2\} = 0, \text{ the solution of which was } \{\xi2\} = \begin{Bmatrix} 0.988 \\ -0.153 \end{Bmatrix}.$$

Thus, the calculated eigenvector $\{\xi2\}$ for mode-2 shows the motion of the rim and backboard to be 180° out of phase, exactly as shown in empirical Figure 6. Furthermore, the ratio of the rim-to-backboard motion for mode 2 was calculated as 0.988/0.153 = 6.45. This calculated ratio of the rim-to-backboard motion of 6.45 was reasonably close to the empirical ratio of rim-to-backboard motion of approximately 7.5, as measured by using mechanical calipers and zooming in on the maximum displacement vectors shown in Figure 6.

These two eigenvectors $\{\xi1\}$ and $\{\xi2\}$ were then checked for orthogonality using the following equation.

$$\{\xi1\}^T \begin{bmatrix} M1 & 0 \\ 0 & M2 \end{bmatrix} \{\xi2\} = 0$$

The above gave 0.929 × M1 × 0.988 − 0.370 × M2 × 0.153 = 0.929 × 1.1 × 0.988 − 0.370 × 17.8 × 0.153 = 0, which shows that the eigenvectors associated with the Hydra-Rib (no shot-clock) in Table 10 were indeed orthogonal.

As a check, these two eigenvectors {*ξ1*} and {*ξ2*} were then transformed to {*v1*} and {*v2*} using the proper orthogonal decomposition (POD), also known as the Karhunen–Loève decomposition, as described by Feeney and Kappagantu.

$$\{\xi\} = \begin{bmatrix} 1/\sqrt{M1} & 0 \\ 0 & 1/\sqrt{M2} \end{bmatrix} \{v\}$$

The eigenvector {*ξ1*} = {0.929, 0.370}$^T$ transformed to {*v1*} = {0.9744, 1.5611}$^T$, and the eigenvector {*ξ2*} = {0.988, −0.153}$^T$ transformed {*v2*} = {−1.0364, 0.6469}$^T$. The resulting dot products {*v1*}$^T$*{*v2*} = {*v2*}$^T$*{*v1*} = 0 show that the POD form of the eigenvectors was also orthogonal.

## 4. Discussion and Conclusions

ERTD readings for the four rim-backboards are also shown in Table 10. The correlation = PEARSON function in Excel gave a cross-correlation coefficient of 95.67% in the inverse correlation between the reading of the Energy Rebound Testing Device and the rim spring rate (stiffness) *K1* from the data listed in Table 10. A graph of the ERTD reading in percent versus rim spring rate *K1* in kN/m is shown in Figure 14. The least-squares equation for the ERTD reading, Figure 14, is as follows:

$$\text{ERTD \%} = 85.7368 - 1.04364 \times K1$$

where the spring rate of *K1* is in kN/m.

Thus, the percentage change in kinetic energy reading of the ERTD increased as the K1 became softer. This meant that the ERTD was a portable and easy-to-use measure of rim stiffness, and it could be used to provide more consistency to the sport of basketball. Too soft a rim (low *K1*) could affect the outcome of a basketball game. Too soft a rim was often indicative of loose mounting bolts that needed corrective tightening, or backboard damage such as hidden cracks. Hidden cracks might be where the rim is attached to the backboard and possibly hidden from sight by the bracket supporting the rim. Cracks in the backboard necessitated replacement of the backboard for safety reasons, as well as consistency in rim stiffness.

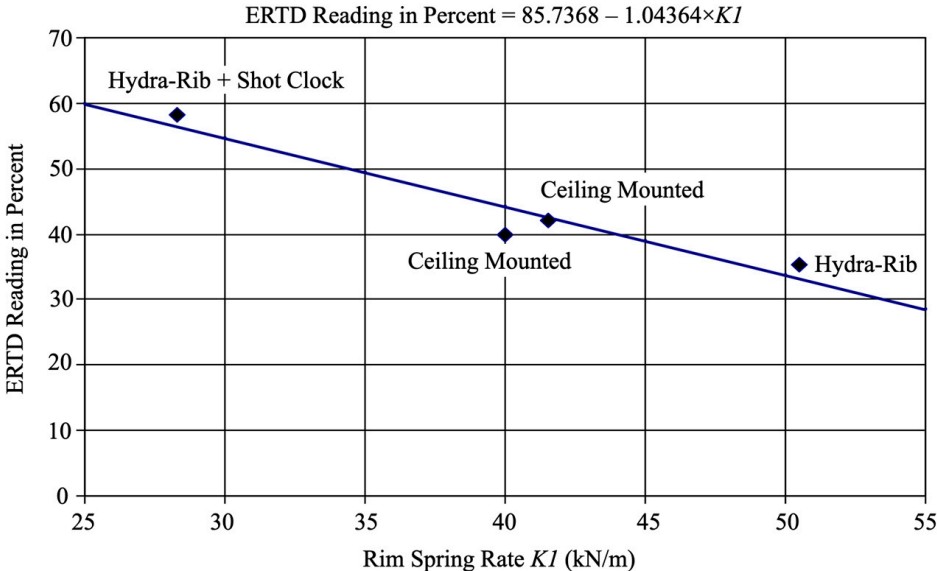

**Figure 14.** ERTD reading in percent versus rim spring rate *K1* in kN/m.

In Article 4 of Section 15, Baskets-Ring of the 2022–2023 NCAA Men's Basketball Rules Handbook [11] states, "All competitive rings shall be tested for rebound elasticity once before the season before the first date of competition and once before the postseason. The

rebound elasticity requirement shall be 35% to 50% energy absorption and within a 5% differential between baskets on the same court". Thus, the Energy Rebound Testing Device could help ensure that the rim (ring) spring rates were similar on both ends of the court, so that as teams switched ends during the basketball game, the rims were consistent.

In conclusion, Figure 14 graphically depicts the cross-correlation coefficient of 95.67% in the inverse correlation between the reading of the Energy Rebound Testing Device and the rim spring rate (stiffness). Thus, we concluded that the kinetic-energy calculations made by the ERTD functioned as intended by the inventors. In order to establish this cross-correlation, we modeled the first two modes of rim-backboard vibration via a two-degree-of-freedom lumped-parameter spring-mass system. By using a known perturbation mass applied to the end of the rim, we were able to define four equations used to ascertain four unknowns: the spring rates and dynamic masses of the rim and backboard. Subsequently calculated eigenvectors were compared favorably with empirically measured eigenvectors for these first two modes of vibration.

**Author Contributions:** Conceptualization, K.P.N.; methodology, D.W.; validation, K.P.N., D.W. and T.W.; formal analysis, D.W. and T.W.; investigation, D.W. and T.W.; writing—original draft preparation, D.W.; writing—review and editing, K.P.N., D.W. and T.W.; visualization, K.P.N. and D.W.; supervision, K.P.N.; project administration, K.P.N. All authors have read and agreed to the published version of the manuscript.

**Funding:** This research received no external funding.

**Data Availability Statement:** The Frequency Response Function *.FRF files became corrupted and are no longer available.

**Conflicts of Interest:** The authors declare no conflict of interest.

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
