# Peer review of "Modes of Vibration in Basketball Rims and Backboards and the Energy Rebound Testing Device"

_vibration, doi:10.3390/vibration6040045_

Round 1
Reviewer 1 Report
This paper is an interesting application of standard modal analysis. None of the methods are novel and the techniques applied are simple.
I wouldn't recommend a work like this one for publication on a research journal but since this is a special issue about applications of vibration theory in sport, it definitively fits the bill and it would be of interest for readers attracted to the title of the issue.
Suggestions to improve the manuscript:
· There is no literature review, can you put your work in context in the introduction?
· The font size on the tables is much bigger than the rest of the text, please use the same font throughout the paper.
· Some of the figures are too big, please resize such that the fonts are the same than the main text (figure 1, figure 3).
· Figures 4 and 5 could be omitted, they are simple screenshots from a signal analyzers.
· Please separate the plot and the equations in figure 15. Put the equations separate and is standard papers, again, keep font size the same.
· Add more depth into the discussion, why is important to know the stiffness of the basketball rims, what methods are used currently, was the simple 2DOF model able to predict accurately the modeshapes and natural frequencies? are these modal results interesting in any other way?
Author Response
This is a summary of the many updates made to the manuscript based on the valued comments of the three reviewers and the editor.
The revised manuscript has DJW added as a suffix in the file name, so that you may distinguish it from what was sent to me for editing.
Update 1: Figures 4 and former Figure 5 are now combined. We authors admit that it was unnecessary to have both Figures 4 and 5. However, we authors please ask that we are able to retain the new Figure 4, as the analyzer settings shown in Figure 4 were crucial to all of the modal analyses which were performed. The primary editor of the vibrations journal formally asked that all information needed by anyone replicating our work be included in our paper, and that is why we ask that Figure 4 please be retained.
Update 2: Former Figure 15 (differential equations of motion) has been removed. The 2DOF spring-mass illustration formerly in Figure 15 has been moved to Figure 13, which is actually a very nice home for it. As requested by one reviewer, the equations in former Figure 15 are now typed in as text, which was also a substantial improvement. This same reviewer wondered if there was more information to be gathered from the mode shapes, which led to the calculation the eigenvectors of the first two modes of vibration. Not only did phase of the calculated eigenvectors agree with the phase of the measured eigenvectors in Figures 5-6, but the calculated and measured ratios of rim-to-backboard motions were reasonably close. Thank you, Thank you, Thank you for leading the authors towards this additional analysis.
Update 3, Added to the end of the Introduction (and to the references): This research complimented the work of others. The Abbott-Davis patent taught the Energy Rebound Testing Device, which was prominent in our study. Javorski, et-al., characterized the dynamic behavior of a ceiling mounted basketball goal. Okubo and Hubbard analyzed the dynamics of basketball-rim interactions. Russell modeled basketballs as spherical acoustic cavities. Thus, our contribution is but one facet of the overall field of vibrations in the sport of basketball.
Please note that Google Scholar, ResearchGate, Google, and a professional searcher were consulted to find the Javorski, Okubo-Hubbard, and Russell references. Every effort was made to make this search comprehensive.
Update 4, Comments on Figure 13 now read: Figure 13 shows the Fair-Court® Energy Rebound Testing Device, which is a long rod having a hook on the upper end which removably fastens to the end of the basketball rim. Along the long rod is a stop, which the drop-mass is held against before the drop-mass makes a 0.76m (30 inch) drop to the base of the ERDT. Within the base of the ERTD is a compression spring, which causes the drop-mass to rebound. Also within the base is a photo-sensor which detects the transit of a 100mm long highly reflective portion of the drop-mass both during its initial descent towards the compression spring, and its subsequent rebound. The 100mm distance ΔZ divided by the downward transit time ΔT1 gave the drop velocity ΔZ/ΔT1. The same 100mm distance ΔZ divided by the rebound transit time ΔT2 gave the rebound velocity ΔZ/ΔT2. The ratio of the change in kinetic energy divided by the original kinetic energy was given by this expression: [m*(ΔZ/ΔT1)2 -m*(ΔZ/ΔT2)2]/[m*(ΔZ/ΔT1)2], where drop-mass m was 0.74 kg. Since drop-mass m and ΔZ2 occurs in both the numerator and denominator, the kinetic energy ratio was simplified to [1/ΔT12–1/ΔT22]/[1/ΔT12], which was further simplified to 1–(ΔT1/ΔT2)2, agreeing precisely with column 12 of the Abbott-Davis patent. This is the reading displayed by the ERTD, and it is a measure of the energy absorbed by the basketball rim and backboard.
Update 5, Discussion and Conclusions: This section title relocated to include the discussion of the Energy Rebound Testing Device, as suggested by one reviewer.
Update 6, Added to the Discussion and Conclusions:
Too soft a rim (low K1) could affect the outcome of a basketball game. Too soft a rim was often indicative of loose mounting bolts which needed corrective tightening, or backboard damage such as hidden cracks. Hidden cracks might be where the rim attached to the backboard and possibly hidden from sight by the bracket supporting the rim. Cracks in the backboard necessitated replacement of the backboard for safety reasons, as well as consistency in rim stiffness.
Figure 14 graphically depicted the cross-correlation coefficient of 95.67% in the inverse correlation between the reading of the Energy Rebound Testing Device and the rim spring rate (stiffness). Thus, we concluded that the kinetic-energy calculations made by the ERTD functioned as intended by the inventors. In order to establish this cross-correlation, we modeled the first two modes of rim-backboard vibration via a two degree-of-freedom lumped-parameter spring-mass system. By use of a known perturbation mass applied to the end of the rim, we were able to define four equations used to ascertain four unknowns, the spring rates and dynamic masses of the rim and backboard. Subsequently calculated eigenvectors compared favorably with empirically measured eigenvectors for these first two modes of vibration.
Figure 14 has been annotated as to which data points were Hydra-Rib + Shot Clock, Ceiling Mount, or Hydra-Rib (without a shot clock). I hope you find this helpful and informative.
Update 7, Acknowledgements removed: Unable to reach the widow of Dr. Jerry Krause, and with major surgery coming very soon, I have removed the acknowledgements section and all mentions of Dr. Krause elsewhere. We authors will go on thanking him in our hearts.
Update 8 (English Language Editing): Once the reviewers and editors have ascertained that the technical content of our manuscript has stabilized, we authors will gladly pursue the English Language editing service offered by MDPI. We definitely want to present our research in the best possible manner. Please do not publish our manuscript until the English Language editing is completed.
General Comment: All tables have been stored as images by the editors of this manuscript. All figures are images, too. If the editors desire smaller fonts in table and figures, all they have to do is shrink these images with their computer mouse. We authors suspect that there will be such re-sizing as the paper reaches its final form.
Many thanks to the work of the reviewers and editors, as they really helped to advance our manuscript. We authors will forever be indebted. If there are any more changes needed, we will gladly make them.
Thanks so very much,
Daniel Winarski, Kip P. Nygren, Tyson Winarski
Reviewer 2 Report
In this paper, modes of vibration in Basketball Rims and Backboards and the energy rebound testing device are discussed. Although the topic is within the journal's scope, the authors need to address the following issues:
1. Introduction section is too poor. You should cite relevant papers in the introduction. At the end of the introduction, you should clearly list the contributions of the paper.
2. The literature review section is missing. You should include relevant papers in the literature section.
3. You should merge the discussion with the experimental results and analysis
4. Conclude the paper in the conclusion with the contributions and results.
5. Very poor list of references, you should update the introduction, and literature sections by citing more relevant papers.
professional editing service needed to fix the grammar issues
Author Response

(The authors gave the same response as above.)

Reviewer 3 Report
Dear Author,
In your research paper, only six relevant references are currently listed. To further strengthen the foundation of your study, I suggest you consider adding more literature support. This will help enrich your literature review section, providing a more comprehensive background and context.
Author Response

(The authors gave the same response as above.)

Round 2
Reviewer 2 Report
In the revised draft, the authors address some issues, but the following issues need to be addressed:
A very poor introduction and literature review sections. Only 10 papers were cited and mostly too old paper. State-of-the-art models need to be included in the literature section or at the end of the introduction.
No comments
Author Response
We hope the reviewers are happy with the following newly added references.
- Feeny, B. F., and Kappagantu, R. “On the Physical Interpretations of Proper Orthogonal Modes in Vibrations. Downloaded from https://www.egr.msu.edu/~feeny/jsv211.pdf.
- Dumond, Patrick, Dominic Monette, Fadi Alladkani, James Akl, and Inès Chikhaoui. "Simplified setup for the vibration study of plates with simply-supported boundary conditions."MethodsX6 (2019): 2106-2117. Downloaded from https://www.sciencedirect.com/science/article/pii/S221501611930247X .
- Gharaibeh, Mohammad A. and Obeidat, Amr M.. "Vibrations Analysis of Rectangular Plates with Clamped Corners"Open Engineering, vol. 8, no. 1, 2018, pp. 275-283. https://doi.org/10.1515/eng-2018-0030, Downloaded from https://www.degruyter.com/document/doi/10.1515/eng-2018-0030/html?lang=en
Based on the negative assessment of the Introduction, that section has been entirely rewritten. The introduction now includes a useful comparison-contrast discussion of the Javorski article. More information is included regarding the assessment of modes 3-6, which was requested by reviewer #1 on his/her first assessment. Other pertinent references are discussed in the Introduction, which is hopefully more of what reviewer #2 was looking for.
Based on the Feeny article, the eigenvectors calculated for the 2-DOF model of the Hydra-Rib in Table 10 (no shot clock), were shown to be orthogonal. A minor point, but worth mentioning.
Dumond’s notation has been used to annotate the captions for modes 3-6 (Figures 7-10). Based on Dumond’s article, Figure 10 (mode 6) was enhanced to show a higher mode of plate vibration for the backboard. We authors sincerely thank the reviewers for their encouragement to dig deeper, as that clearly brought results regarding Figure 10. Thank you, Thank you, Thank you!
Please forgive this request, but the Abbott-Davis patent (1992) is THE patent on the Energy Rebound Testing Device, which was a major focus of this study. Thus, this reference might be considered “old” by one reviewer, but we authors want to please be allowed to continue to use this reference. It should also be noted that no prior art references by other authors has been found regarding the modes of vibration of the elastic basketball rim and backboard.
We authors thank our reviewers for their patience and valued comments. We will gladly address any and all future changes coming from this review.
Reviewer 3 Report
Overall, your research paper currently includes only 10 relevant references, which might be slightly insufficient. In order to further enhance the foundation of your study, I sincerely recommend considering the addition of more literature support. This will contribute to enriching your literature review section, providing a more comprehensive background and context. Incorporating additional relevant research will ensure that your paper possesses greater depth and breadth within the current research domain.
Author Response

(The authors gave the same response as above.)

Round 3
Reviewer 2 Report
Dear Authors,
Thank you for addressing the issues that I already identified. The authors should still fix the following issues:
1. The list of reference papers is not cited inside the paper. Please cite the reference paper with in the relevant text.
2. Still no recently published relevant papers (like those published within the last 2/3years) are cited in the background study.
Its okay for me
Author Response
These 8 additional references have been added to our paper, and all 21 references are now cited within the body of our paper as they are encountered. The DOI is included, where available, for all references. Two of these new references are dated 2023.
1.Irvine, Tom. “Unit 30: Rectangular Plate Shock & Vibration.” Dynamic Concepts, Inc., page 28, September 29, 2014. Downloaded from https://drive.google.com/file/d/0Bw1VlHh5X_tTT1ZsRnBSUnlUSGs/view?resourcekey=0-ZJS96GaZbys5gCcIdmeW4w 2.AnÄ‘elić, N., M. ÄŒanaÄ‘ija, and Z. Car. "Determination of Natural Vibrations of Simply Supported Single Layer Graphene Sheet using Non-Local Kirchhoff Plate Theory." In IN-TECH 2017 International Conference on Innovative Technologies. Page 5, 2017. 3.Guguloth, Ganesh Naik, Baij Nath Singh, and Vinayak Ranjan. "Free vibration analysis of simply supported rectangular plates." Vibroengineering Procedia 29 (2019): 270-273. DOI: 10.21595/vp.2019.21135. Downloaded from:https://www.researchgate.net/publication/337605921_Free_vibration_analysis_of_simply_supported_rectangular_plates#fullTextFileContent. 4.Okubo, Hiroki, and Mont Hubbard. "Identification of basketball parameters for a simulation model." Procedia Engineering 2, no. 2 (2010): 3281-3286. 5.Agustinus Oey (2023). Vibration of rectangular clamped thin plate. MATLAB Central File Exchange. Downloaded from https://www.mathworks.com/matlabcentral/fileexchange/28375-vibration-of-rectangular-clamped-thin-plate. 6.Model 3500 Positive Lock Breakaway Goal. Updated January 21, 2010. Gared Holdings, LLC. 7.Kuttner, T., Rohnen, A. Experimental Modal Analysis. In: Practice of Vibration Measurement. Springer, Wiesbaden, 2023, pp 513-542, https://doi.org/10.1007/978-3-658-38463-0_15. 8.Getting Started with Modal Curve Fitting, Sep 21, 2020, Siemens. Downloaded from:https://community.sw.siemens.com/s/article/getting-started-with-modal-curvefitting.Figure 12 had its eigenvector arrows reduced in size, slightly.
Many thanks again for the patient and invaluable leadership of the reviewers. If more changes are needed, they need only to ask.
Thanks so very much!
Reviewer 3 Report
After the author's efforts, the article has significantly improved. However, overall, your research paper currently contains only 13 relevant references, which may be somewhat insufficient. To further strengthen the foundation of your research, I sincerely suggest that you consider adding more literature support. This will help enrich your literature review section, providing a more comprehensive background and context. Including additional relevant studies will ensure that your paper has greater depth and breadth in the current research field. I hope that your literature section can increase to over 20 references, especially in the Discussion and Conclusions sections where more references are needed to support your findings.
Here are some suggestions that might help you expand the literature section:
- Similar Studies: Look for other research related to your research topic, especially those that support or complement your claims. You can search for these references in academic journals, research reports, and theses.
- Latest Research: Ensure that you include the latest research findings to reflect the current trends and developments in the field. Regularly check academic journals and online resources for new literature.
- Diverse Perspectives: Include literature that supports different viewpoints. This will help you provide a more comprehensive discussion and comparison in the Discussion section.
- Theoretical Framework: If your research is based on a specific theory, make sure to cite relevant literature related to that theory and explain how your study relates to it.
- Empirical Studies: If your research involves empirical data, cite literature from similar studies to support your methods and results.
Finally, make sure to list all references in the appropriate citation format in the literature section to ensure that your paper meets academic requirements. I hope these suggestions help you improve your paper's literature section.
Author Response

(The authors gave the same response as above.)
